# Microbiome and Microbial Pure Culture Study Reveal Commensal Microorganisms Alleviate *Salmonella enterica* Serovar Pullorum Infection in Chickens

**DOI:** 10.3390/microorganisms12091743

**Published:** 2024-08-23

**Authors:** Jianshen Zhu, Jinmei Ding, Kaixuan Yang, Hao Zhou, Wenhao Yang, Chao Qin, Liyuan Wang, Fuquan Xiao, Beibei Zhang, Qing Niu, Zhenxiang Zhou, Shengqing Yu, Qizhong Huang, Shaohui Wang, He Meng

**Affiliations:** 1Shanghai Key Laboratory of Veterinary Biotechnology, Department of Animal Science, School of Agriculture and Biology, Shanghai Jiao Tong University, Shanghai 200240, China; zhujianshen@sjtu.edu.cn (J.Z.); dingjinmei@sjtu.edu.cn (J.D.); zhouhao1992@sjtu.edu.cn (H.Z.); yangwenhao@sjtu.edu.cn (W.Y.); qin_chao@sjtu.edu.cn (C.Q.); kkkly88@sjtu.edu.cn (L.W.); xiaofuquan@sjtu.edu.cn (F.X.); 2Animal Husbandry and Veterinary Research Institute, Shanghai Academy of Agricultural Science, Shanghai 201403, China; yangkaixuan007@hotmail.com (K.Y.); nqdwkx@163.com (Q.N.); zhouzhenxiang@hotmail.com (Z.Z.); huangqizh@163.com (Q.H.); 3Shanghai Veterinary Research Institute, the Chinese Academy of Agricultural Sciences, Shanghai 200241, China; bzhang1997@sina.com (B.Z.); yus@shvri.ac.cn (S.Y.)

**Keywords:** pullorum disease, *Salmonella pullorum*, chicken gut microbiota, *Lacticaseibacillus*

## Abstract

Pullorum disease, an intestinal disease in chickens caused by *Salmonella enterica* serovar pullorum (*S.* Pullorum), is a significant threat to the poultry industry and results in substantial economic losses. The bacteria’s transmission, both vertical and horizontal, makes it difficult to completely eliminate it. Control strategies for pullorum disease primarily involve stringent eradication programs that cull infected birds and employ antibiotics for treatment. However, eradication programs are costly, and antibiotic use is restricted. Therefore, developing alternative control strategies is essential. Increasingly, studies are focusing on modulating the gut microbiota to control intestinal diseases. Modulating the chicken gut microbiota may offer a novel strategy for preventing and controlling pullorum disease in poultry. However, the impact of *S.* Pullorum on the chicken gut microbiota has not been well established, prompting our exploration of the relationship between *S.* Pullorum and the chicken gut microbiota in this study. In this study, we initially analyzed the dynamic distribution of the gut microbiota in chickens infected with *S.* Pullorum. Alpha diversity analysis revealed a decrease in observed OTUs and the Shannon diversity index in the infected group, suggesting a reduction in the richness of the chicken gut microbiota due to *S.* Pullorum infection. Principal coordinate analysis (PCoA) showed distinct clusters between the gut microbiota of infected and uninfected groups, indicating *S.* Pullorum infection changed the chicken gut microbiota structure. Specifically, *S.* Pullorum infection enriched the relative abundance of the genera *Escherichia-Shigella* (65% in infected vs. 40.6% in uninfected groups) and *Enterococcus* (10.8% vs. 3.7%) while reducing the abundance of *Lactobacillus* (9.9% vs. 32%) in the chicken microbiota. Additionally, based on the observed changes in the chicken gut microbiota, we isolated microorganisms, including *Bifidobacterium pseudolongum*, *Streptococcus equi* and *Lacticaseibacillus paracasei* (*L. paracasei*)*,* which were decreased by *S.* Pullorum infection. Notably, the *L. paracasei* Lp02 strain was found to effectively inhibit *S.* Pullorum proliferation in vitro and alleviate its infection in vivo. We found that *S.* Pullorum infection reduced the richness of the chicken gut microbiota and enriched the relative abundance of the genera *Escherichia-Shigella* and *Enterococcus* while decreasing the abundance of the anaerobic genus *Lactobacillus.* Furthermore, microbiota analysis enabled the isolation of several antimicrobial microorganisms from healthy chicken feces, with a *L. paracasei* strain notably inhibiting *S.* Pullorum proliferation in vitro and alleviating its infection in vivo. Overall, this research enhances our understanding of the interaction between gut microbiota and pathogen infection, as well as offers new perspectives and strategies for modulating the chicken gut microbiota to control pullorum disease.

## 1. Introduction

Pullorum disease, an illness that selectively afflicts chickens due to *Salmonella enterica* serovar pullorum (*S.* Pullorum), is characterized by symptoms such as intestinal white diarrhea [1,2,3]. Additionally, horizontal and vertical transmission of *S.* Pullorum can occur from infected chickens to their counterparts and offspring via various media and egg, respectively, which make it difficult to completely eliminate the bacteria [4,5,6,7]. The disease continues to cause significant economic losses due to the limitations of the current control measures [8]. Current pullorum disease control strategies primarily involve strict eradication through culling infected birds and antibiotic treatment [9,10,11,12]. However, the strict eradication program is costly [13,14], and antibiotic usage is restricted [15,16]. Thus, developing alternative prevention and treatment strategies for pullorum disease is crucial.

The gut microbiota are crucial for maintaining the overall health and physiological functions of the host [17,18,19,20]. Increasingly, research is focusing on the interaction between the gut microbiota and gastrointestinal disorders [21,22,23,24,25]. Studies have linked changes in the gut microbiota composition and diversity to the development and progression of gastrointestinal disorders. Research on recurrent *Clostridioides difficile* (*C. difficile*) infection (rCDI) has identified gut microbiota dysbiosis as a contributing factor and has shown that fecal microbiota transplantation (FMT) can successfully treat and prevent the disease [26,27,28]. Evidence suggests that FMT reestablishes the gut microbiota, providing colonization resistance to *C. difficile* expansion and transmission [29,30,31]. Similarly, studies on pig gut microbiota have shown that gassericin A, a bacteriocin produced by *Lactobacillus* species from the gut, binds to intestinal epithelial cells, conferring resistance to diarrhea. Collectively, it is clear that modulating the gut microbiota may offer a novel strategy for controlling intestinal diseases.

In the poultry field, numerous studies have explored the link between chicken gut microbiota and production performance. However, the interaction between chicken gut microbiota and gastrointestinal pathogen disorders, particularly *S.* Pullorum infection, remains poorly understood. Understanding the complex interplay between gut microbiota and host gastrointestinal disorders is essential for formulating health-enhancing strategies. Thus, we developed an *S.* Pullorum-infected chicken model for gut microbiota analysis. Here, we report the changes in the gut microbiota following *S.* Pullorum infection. Subsequently, based on microbiota analysis, we isolated microorganisms associated with *S.* Pullorum infection and evaluated their ability to inhibit its proliferation in vitro and alleviate infection in vivo. Overall, this study offers fundamental insights into the interaction between the gut microbiota and *S.* Pullorum, as well as presents a new perspective and strategy for the prevention and treatment of pullorum disease.

## 2. Materials and Methods

### 2.1. S. Pullorum Challenge and Sampling

The chickens employed in the *S.* Pullorum challenge were the offspring of Xin Pudong chickens negative for pullorum disease based on the whole blood plate agglutination test at the Animal Husbandry and Veterinary Research Institute of Shanghai Academy of Agricultural Sciences. The standard strain *Salmonella pullorum* 1218 from the Shanghai Veterinary Research Institute of Chinese Academy of Agricultural Sciences was selected as the challenge strain [32,33]. All animal experiments were conducted in accordance with the Laboratory Animal Research (ILAR) guide of Shanghai Jiao Tong University, China. According to the challenge dose obtained by the LD_50_ test (LD_50_ = 6.3 × 10^6^ CFU, previously measured), 504 newborn chicks were randomly divided into two groups. Among them, 353 Xin Pudong chicks were injected with a PBS suspension of *S.* Pullorum in the leg muscle at a dose of 0.2 mL per chick (group SP), and 151 chicks were injected with the same dose of PBS (group Ctrl) in the same way. All chickens were fed ad libitum and raised without antibiotics and vaccines. The body weight of the chickens was measured, and the *S.* Pullorum in their feces was detected on days 1, 7, 14 and 21 d.p.i. (days post-infection). The death and survival of chickens were counted every day, and the mortality and survival rates were calculated at 21 d.p.i. The liver and spleen at 21 d.p.i. were collected for the organ index and detection of *S.* Pullorum. For the detection of *S.* Pullorum in feces and organs, we used small steel balls to shake and grind these samples into liquids and then cultivated *S.* Pullorum using the selective medium BSA. The cecal contents at 14 d.p.i. were collected for gut microbiota analysis (Appendix A), and then, the samples were stored in a −80 °C refrigerator for further use.

### 2.2. 16S rRNA Gene Amplicon Sequencing

Fecal contents of the chicks were collected at 14 days post-infection for gut microbial genome sequencing, and 60 and 48 fecal samples were collected from infected and uninfected chickens, respectively. The fecal genome DNA extraction kit (Cat# DP328, Tiangen, China) was used to extract the gut microbial DNA. The V3–V4 hypervariable region of the gut microbial 16S rRNA gene was amplified and sequenced using the NovaSeq PE250 platform (Illumina, San Diego, CA, USA). The Quantitative Insights Into Microbial Ecology 2 (QIIME2, version 2019.4) pipeline was employed to process the sequencing data, including primers removing, denoise, splice, chimeras deletion and dereplication. The sequence obtained after quality control is called ASVs (amplicon sequence variants), corresponding to the operational taxonomic units (OTUs). Combine the ASVs feature sequence with the ASV table and remove singleton ASVs (i.e., ASVs with only 1 sequence in the total sample). After the ASV feature sequence is obtained, the length distribution of the high-quality sequences contained in all the samples is counted.

### 2.3. Species Annotation and Diversity Analysis

Silva database was used to compare and annotate 16S rRNA genes of the bacteria. To minimize the difference in sequencing depths across samples, it is necessary to obtain an averaged, rounded rarefied OTU abundance table. We randomly selected a certain number of sequences from each sample, which is the sparse method, to ensure that all samples reached the same depth. Then, the observed OTU and its relative abundance of each sample at this sequencing depth were predicted. For this evenly resampled OTU table, we analyzed and obtained the number of taxa contained in each sample at each taxonomic level (domain, phylum, class, order, family, genus and species). Then, the column chart was used to show the taxonomic composition of the microbial community. The alpha of the microbial community was analyzed based on the OTU abundance table. The indexes of Shannon represent the microbial community’s richness. The differences in gut microbial beta diversity can be displayed by the principal coordinates analysis (PCoA).

### 2.4. S. Pullorum Infection Related Microorganisms Isolation and Identification

These strains (including *Lactobacillus*, *Pediococcus*, *Lactococcus*, *Clostridium*, *Bifidobacterium* and *Streptococcus*) were isolated from the feces of chickens directly using the corresponding selective medium to investigate the microorganisms associated with resistance to *S. pullorum* infection. Anaerobes were incubated with anaerobic tanks and deaerants for anaerobic culture. The MRS (de Man, Rogosa and Sharpe) medium combined with the BL medium (Bifidobacterium Lactobacillus medium) was used for the isolation of *Lactobacillus* spp., *Pediococcus* spp. and *Bifidobacterium* spp. The reinforced Clostridial agar medium was for *Clostridium* spp., Elliker agar medium for *Lactococcus* spp. and the bile Esculin agar medium and Columbia blood agar were combined for *Streptococcus* spp. These mentioned media were produced by Hope Bio-Technology Co., Ltd. (Qingdao City, Shandong Province, China). The pure cultures obtained were subjected to full-length 16S rDNA gene sequencing for microbial identification. Additionally, mix the bacterial suspension with 50% sterile glycerol in a 1:1 ratio and store it in a −80 °C freezer for long-term preservation (Appendix A).

### 2.5. Antimicrobial Test

The agar diffusion assay was used to test the antimicrobial activity. The indicator strain *S.* Pullorum (200 μL, 10^8^ CFU/mL) was spread on 20 mL LB agar in a Petri dish (9 cm), and 200 μL cell-free supernatant of isolated microorganisms was added into the Oxford cup (a stainless cylinder, outer diameter 7.8 ± 0.1 mm, inner diameter 6.0 ± 0.1 mm and height 10.0 ± 0.1 mm) placed on the surface of the agar. Then, the Petri dish was incubated at 37 °C for 18 h. The size of the clear diffusion zone around the cup (including that the Oxford cup was 7.8 mm) was measured with a vernier caliper and reported in millimeters (mm).

### 2.6. S. Pullorum Challenge–Intervention Animal Experiment

After verifying *Lacticaseibacillus paracasei* (*L. paracasei*) Lp02 antibacterial ability in vitro, we orally administered it to chicks (specific pathogen-free chickens, White Leghorn breed, from Jinan city, Shandong Province, China, SAIS Poultry Co., Ltd.) to measure its protective ability in vivo against *S.* Pullorum infection, so 90 newly hatched chicks were randomly divided into three groups, each consisting of 30 chicks. In the challenge with intervention group (group Infect_LP), the hatched chicks freely drank water containing 10^8^ CFU/mL of *L. paracasei* Lp02 for 3 days and were orally challenged with 0.2 mL, 10^9^ CFU/mL *S.* Pullorum at 4 days old. In the challenge without intervention group (group Infect), the chicks received no treatment during the first three days and were challenged orally with *S.* Pullorum in the same way at 4 days old. The control group (Ctrl) received no treatment. At 1, 7 and 14 d.p.i., the body weight was measured, and 12 fecal samples at each time point from each group were collected for the detection of *S.* Pullorum.

### 2.7. Metabolite Detection

To demonstrate the underlying mechanism behind *L. paracasei* Lp02 (LP_Pos) strain and *L. paracasei* TCAN14 having different antibacterial capabilities, we examined their metabolite compositions. Six single colonies of the *L. paracasei* Lp02 (LP_Pos) strain and six single colonies of the *L. paracasei* TCAN14 (LP_Neg) strain, each derived from a single clonal strain, were picked and inoculated into separate LB broth and incubated overnight at 37 °C with shaking at 180 rpm. After the overnight growth, the cultures were centrifuged at 4000 rpm for 10 min at 4 °C. The supernatants were filtered through 0.22 μm filters to obtain sterile supernatants. The supernatants were immediately frozen in liquid nitrogen and stored at −80 °C until further analysis. Then, metabolome was extracted from the gut content sample through the utilization of methanol, and a quality control (QC) sample was fabricated by combining aliquots of each sample. The acquisition of the metabolic profile was executed on a Q-Exactive mass spectrometer (Thermo Scientific, Waltham, MA, USA) equipped with an electrospray ionization (ESI) source. The mass spectrometer was interconnected with a Waters ACQUITY UPLC system (Waters, Milford, MA, USA). Chromatographic separation was carried out on an ACQUITY UPLC HSS T3 column (1.8 μm, 100 mm × 2.1 mm i.d.) (Waters, Milford, MA, USA). The normalized data were submitted to SIMCA-P software (version 13.0, Umetrics, Uppsala, Sweden) for principal component analysis (PCA) and orthogonal partial least squares discriminant analysis (OPLS-DA). A 999-time permutation test was conducted to validate the developed OPLS-DA models. Based on accurate mass measurement, metabolites identification with significant alterations were searched against the Human Metabolome Database (HMDB; http://www.hmdb.ca/ (accessed on 10 July 2024)) with a 10-ppm molecular weight tolerance. Moreover, the product ion spectrum of the metabolite was matched with MS spectra accessible in HMDB to affirm the identification. The identified biomarkers were subjected to MetaboAnalyst 4.0 (https://www.metaboanalyst.ca (accessed on 10 July 2024)) for the pathway enrichment analysis.

### 2.8. Data Statistics

A Mann–Whitney test was used to test differences and was performed in GraphPad Prism (version 10.0). PERMANOVA test was performed on the Personal genes cloud platform (https://www.genescloud.cn/home (accessed on 15 June 2024)). In the figures, data were presented with the mean ± SD. There was a statistical significance when *p* < 0.05 (* *p* < 0.05, ** *p* < 0.01 and *** *p* < 0.001).

## 3. Results

### 3.1. The Impact of S. Pullorum Challenge on Chicken Performance and Physiology

To study the impact of *S.* Pullorum on the gut microbiota of chickens, we first established an animal infection model by inoculating *S.* Pullorum to chicks. After the *S*. Pullorum challenge, *S.* Pullorum-infected chickens (group SP) showed a significant decrease in body weight at 7, 14 and 21 d.p.i., with average losses of 14.2 g, 17.3 g and 17.4 g, respectively, compared to the non-infected chickens (group Ctrl), which were 109.5 g, 154.7 g and 171.2 g, respectively (Figure 1a). Moreover, the 21-day mortality rate in group SP was 25.63%, exceeding that of group Ctrl at 11.9%, and in the SP group of 353 chicks, there were 91 deaths, compared to 16 in the Ctrl group of 151 chicks (Figure 1b). The deaths peaked at 5 d.p.i., with no further mortality observed by 10 d.p.i. (Appendix A). *S.* Pullorum was also detected in the feces, kidneys and livers of the SP group, contrasting with the absence in the Ctrl group (Appendix A). The livers of the SP group exhibited grayish-white nodules, indicative of pullorum disease infection (Appendix A). At 21 d.p.i., the livers and spleens of the SP group chicks were enlarged, with elevated organ indices (organ weight/body weight × 100) relative to the Ctrl group (Figure 1c). In summary, these results confirm the successful establishment of the infected chicken model, suitable for further gut microbiota analysis.

### 3.2. S. Pullorum Infection Altered the Richness and Structure of Chicken Gut Microbiota

Measurement of the within-sample diversity (α-diversity) using the observed OTUs and the Shannon diversity index revealed a significant difference between the *S.* Pullorum-infected chicken (group SP) and the non-infected chicken (group Ctrl). Group SP exhibited lower gut microbiota diversity compared to the group Ctrl, as depicted in Figure 2a, suggesting a reduction in microbial richness due to infection. Principal coordinate analysis (PCoA) using Bray–Curtis distances demonstrated distinct clustering between the SP and Ctrl groups (Figure 2b), with separation along the primary axis, signifying that *S.* Pullorum infection was the predominant driver of gut microbiota variation (*p* < 0.05 by PERMANOVA). This significant variation was also detectable at the phylum level. In group SP, *Firmicutes* were more abundant relative to the Ctrl, while *Proteobacteria* showed the opposite trend (Figure 2c). Overall, these findings qualitatively indicate that *S.* Pullorum infection significantly altered the chicken gut microbiota.

To identify the gut microorganisms associated with *S.* Pullorum infection, we initially compared the relative abundance of genera between the infected and non-infected chicken gut microbiota. The results showed differences in the abundance of 14 genera (with relative abundances > 0.01%) between the two groups (Table 1). Among the most abundant genera, *Escherichia-Shigella* and *Enterococcus* were more prevalent in infected chickens, whereas *Lactobacillus* was less so (Figure 2d and Table 1). Consistent with this, LEfSe analysis (with LDA scores > 2) identified *Escherichia-Shigella* and *Enterococcus* as the biomarkers of *S*. Pullorum-infected chickens and *Lactobacillus* as a biomarker of the non-infected chickens. Additionally, it is noteworthy that *Streptococcus*, *Ruminococcus*, *Ralstonia, Clostridium, Butyricicoccus, Clostridium, Caulobacter* and *Lactococcus,* which were less abundant, also serve as biomarkers for *S.* Pullorum-infected chickens (Figure 2e). In summary, these decreased microorganisms could be associated with resistance to *S.* Pullorum infection.

### 3.3. Infection Decreased Microorganisms Isolated from Healthy Chicken Feces and Inhibited S. Pullorum Proliferation

The microbiome analysis revealed *S.* Pullorum infection led to a decrease in the abundance of the genus represented by *Lactobacillus* in the chicken microbiota. These genera may be associated with resistance to *S.* Pullorum infection. To validate this, we isolated related microorganisms from the feces of healthy 14-day-old chickens. A total of 106 microbial strains, derived from 48 fecal samples, were isolated on various media, including blood agar, MRS and RCA media (Appendix A). After eliminating duplicates, 38 unique strains were identified, comprising 29 *Lactobacillus* spp.; 4 *Pediococcus* spp.; 2 *Lactococcus spp*. and 1 each of *Clostridium*, *Bifidobacterium* and *Streptococcus* spp. (Appendix A). Using the agar diffusion method for in vitro antibacterial testing, we found that specific isolates of *Bifidobacterium pseudolongum* and *Streptococcus equi* showed antibacterial activity against *S.* Pullorum, with inhibition zones of 9 ± 0.3 mm and 8.7 ± 0.1 mm, respectively (Appendix A). Notably, an isolated strain of *Lacticaseibacillus paracasei* (*L. paracasei* Lp02) demonstrated significantly superior inhibitory effects on *S.* Pullorum growth, with an inhibition zone of 28.2 ± 2.4 mm, compared to another strain, *L. paracasei* TCAN14, which showed an inhibition zone of 9.1 ± 0.4 mm (Figure 3a and S3e). Consequently, we selected *L. paracasei* Lp02 for further investigation.

### 3.4. A Lacticaseibacillus paracasei Isolated from Chicken Feces Alleviated S. Pullorum Infection

To determine whether *L. paracasei* Lp02 plays a protective role against *S.* Pullorum infection in vivo, a challenge–intervention animal experiment was carried out. We orally inoculated newborn chicks with *L. paracasei* Lp02 for 3 days and then challenged them with *S.* Pullorum (Figure 3b). After 14 days post-infection (d.p.i.), chickens infected with *S.* Pullorum and intervened with *L. paracasei* Lp02 (group Infect_LP) harbored 3.46 × 10^2^ CFU/g feces, while chickens without intervention (group Infect) harbored 4.89 × 10^5^ CFU/g feces. These suggested that Infect_LP chickens shed significantly less *S.* Pullorum in the feces 14 d.p.i. (Figure 3c). Consistent with this observation, the proportion of chickens shedding *S.* Pullorum during infection was significantly different between chickens with different treatments; while 88.89% of Infect chickens shed *S.* Pullorum in the feces, only 58.3% of Infect_LP chickens shed detectable levels of the pathogen (Figure 3d). In addition, the loss in body weight was improved in Infect_LP chickens (Figure 3e). These observations suggest that *L. paracasei* Lp02 could effectively inhibit *S.* Pullorum proliferation in vitro and alleviate its infection in vivo.

### 3.5. Comparative Metabolomic Analysis of Sterile Supernatants from L. paracasei Strains

Untargeted metabolomic analysis was conducted on the sterile supernatants from the *L. paracasei* Lp02 (LP_Pos) and *L. paracasei* TCAN14 (LP_Neg) strains to characterize and compare their metabolic profiles comprehensively. A total of 1623 metabolites were detected and identified. The OPLS-DA analysis (Figure 4a) showed distinct separation between the metabolite profiles of LP_Pos and LP_Neg, indicating significant differences in their global metabolite compositions. Additionally, 212 metabolites showed significant differences between the two strains (*p* < 0.05), with 44 metabolites featuring a VIP value > 1 (Figure 4b). Among these, in strain LP_Pos, 39 metabolites were upregulated, including palatinose, visnagin, DL-phenylalanine and others. In strain LP_Neg, five metabolites were upregulated, including NCGC00347708-02, Pro-Val, Arg-Arg-Lys, tetradifon and butylated hydroxytoluene, and others (Figure 4c). These 44 differential metabolites are primarily associated with arginine and proline metabolism, pyrimidine metabolism and purine metabolism (Figure 4d). Notably, the differential levels of five key metabolites—trimethoprim, triacetic acid lactone, fluquinconazole, kasugamycin and chenodeoxycholate—were analyzed and presented. Overall, these results indicate significant differences in the metabolic compositions of the two strains, with the differential metabolites, particularly the five key ones, potentially contributing to the antibacterial capabilities.

## 4. Discussion

In this study, we explored the effects of *S.* Pullorum infection on the chicken gut microbiota, and the role of *S.* Pullorum infection decreased commensal microorganisms in inhibiting *S.* Pullorum, with a focus on developing alternative strategies for the control of pullorum disease. Our findings offer new perspectives on the relationship between *S.* Pullorum and the gut microbiota, specifically within the context of pullorum disease.

Infection with *S.* Pullorum induces changes in the chicken gut microbiota, including a reduction in richness and alterations in the composition. These findings align with our previous research on the gut microbiota changes in pullorum-positive chickens on the farm. The gut microbiota of pullorum-positive chickens on the farm (groups SP_farm and Ctrl_farm) differed from that of laboratory-induced *S.* Pullorum-infected chickens (groups SP_Lab and Ctrl_Lab) due to environmental and age variations, yet the distinction between the microbiota of the infected and uninfected chickens remained consistent across both settings (Appendix A). Comparative analysis of the relative abundance and LEfSe analysis revealed that *Escherichia-Shigella* and *Enterococcus* were significantly more abundant while *Lactobacillus* was less so in pullorum-positive chickens (Appendix A). Reports indicate that *S.* Pullorum infection can provoke inflammatory responses within the chicken intestine and establish its ecological niche [34,35,36]. This pathogen facilitates its expansion by modifying the intestinal environment and utilizing specialized respiratory and metal acquisition mechanisms, outcompeting other commensal microorganisms [37,38]. This benefits gut microorganisms, like *Escherichia-Shigella* and *Enterococcus*, that share a similar ecological niche with *S.* Pullorum, increasing their prevalence [39]. Conversely, other gut microorganisms, exemplified by *Lactobacillus*, that lack an advantage in the inflammation-altered microecology, see a significant reduction in their numbers [40], leading to reduced richness of the chicken gut microbiota. In contrast, the reduction of certain gut microorganisms due to *S.* Pullorum infection could signify their antagonistic nature towards the pathogen, suggesting their potential importance in resisting *S.* Pullorum and their use in microbial intervention strategies for infection.

Microbiome studies can assess gut microbiota diversity and identify potential biomarkers associated with host health and disease. However, obtaining microorganism strains through pure culture is fundamental for functional verification and subsequent microorganism-based interventions [41,42,43]. In this study, based on the gut microbiota results, we isolated the microorganisms which abundance were reduced by *S.* Pullorum infection. Utilizing various culture media, we isolated 39 microbial strains, including *Lactobacillus*, *Bifidobacterium* and *Streptococcus* species, but were unable to successfully culture other reduced gut microorganisms (Table 1). It is reported that culturing many gut microorganisms is challenging. The difficulty in culturing gut microbiota is multifaceted. Existing media cannot satisfy the complex nutritional requirements, and their survival often relies on symbiosis, and the precise oxygen supply is challenging. Stringent conditions such as temperature, pH and osmotic pressure pose obstacles, while low abundance complicates detection and culturing and slow growth further increases the difficulty [44,45]. Nonetheless, culturomics, through diverse culture conditions and rapid identification, has accelerated the cultivation of numerous novel microorganisms related to human health and disease, providing new insights into host–microorganism interactions [46,47,48].

The gut microbiota mediates colonization resistance against pathogens through indirect and direct mechanisms [49]. Indirectly, the microbiota interacts with the host immune system, thereby altering the epithelial and immune cells to promote antipathogen immunity [50]. Direct mechanisms involve competition for niches and nutrients, contact-dependent killing and the production of antagonistic molecules [51,52]. Collectively, these colonization resistance mechanisms offer protection to the host against enteric bacterial pathogens [53]. In this study, we found that a *L. paracasei* Lp02 strain could effectively inhibit *S.* Pullorum proliferation in vitro and alleviate its infection in vivo. Then, through the metabolites analysis of two *L. paracasei* strains with different capabilities of inhibiting *S.* Pullorum, we revealed some potential antagonistic molecules against *S.* Pullorum proliferation in vitro, including trimethoprim, triacetic acid lactone, fluquinconazole, kasugamycin and chenodeoxycholate. However, the bacteriostatic functions of these metabolites require further validation. Meanwhile, how *L. paracasei* Lp02 alleviates *S.* Pullorum infection in vivo still needs to be explored for manipulating the gut microbiota to potentially manage other pathogen-related diseases.

Overall, our study advances the understanding of the gut microbiota’s role in *S.* Pullorum infection and suggests a microbiota-based strategy for preventing and treating pullorum disease. Nonetheless, additional research into the antibacterial mechanisms of *L. paracasei* Lp02 is essential for developing a more comprehensive and effective approach to controlling pullorum disease and other poultry gastrointestinal disorders.

## 5. Conclusions

In this study, we initially characterized the alterations in the gut microbiota following the *S.* Pullorum challenge, identified microorganisms linked to the infection and isolated a *L. paracasei* strain that inhibits *S.* Pullorum proliferation in vitro and ameliorates infection in vivo. Meanwhile, analysis of its metabolic products provided insights into its potential antibacterial substances. Given the global ban on antibiotics and the limitations of traditional measures, developing alternative strategies for *S.* Pullorum control is essential. Modulating the chicken gut microbiota presents a novel approach for managing salmonellosis and other infectious diseases.

## Figures and Tables

**Figure 1 microorganisms-12-01743-f001:**
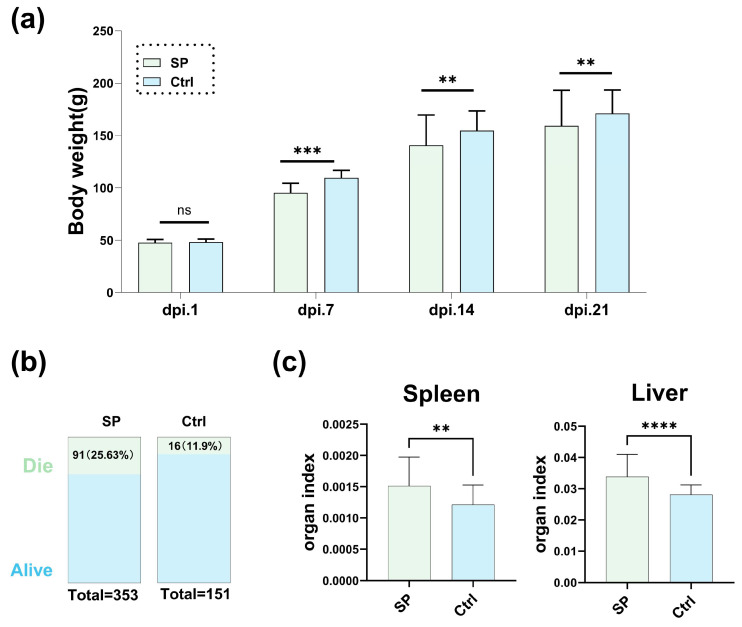
Effect of *S.* Pullorum challenge on chicken performance and physiology. (**a**) Body weights of challenged and control chicks at 1, 7, 14 and 21 days post-infection (d.p.i.). (**b**) Survival and mortality percentages of the two groups at 21 d.p.i. (**c**) Organ indices (calculated as (organ weight/body weight) × 100 of the corresponding chick) of the two groups at 21 d.p.i. **: *p* < 0.01, ***: *p* < 0.001, ****: *p* < 0.0001, ns: not significant and *p* > 0.05.

**Figure 2 microorganisms-12-01743-f002:**
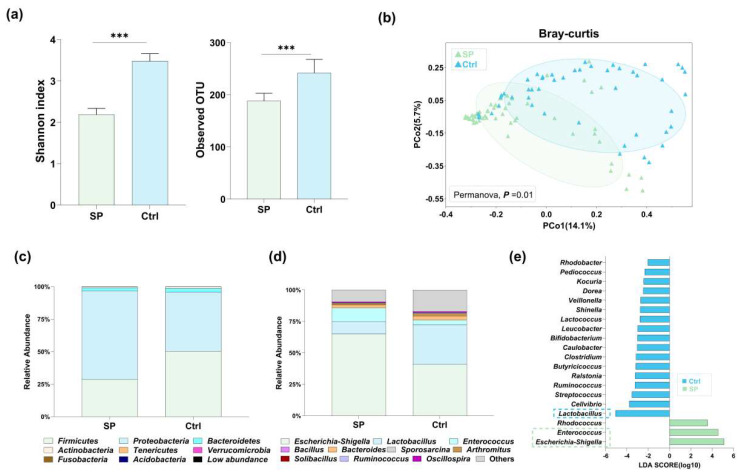
Gut microbiota alterations in chickens following the *S.* Pullorum challenge. (**a**) Comparison of the alpha diversity between the *S.* Pullorum-infected and uninfected groups using the Shannon index. (**b**) Principal coordinate analysis (PCoA) of the Bray–Curtis distance revealed the gut microbiota of *S.* Pullorum-infected and uninfected chickens. Relative abundance of the gut bacterial taxonomic compositions at the phylum (**c**) and genus levels (**d**). (**e**) LEfSe analysis to identify the key biomarkers of chicken gut microbiota, the blue dashed frame represents the Ctrl group and the green dashed frame indicates the SP group. ***: *p* < 0.001.

**Figure 3 microorganisms-12-01743-f003:**
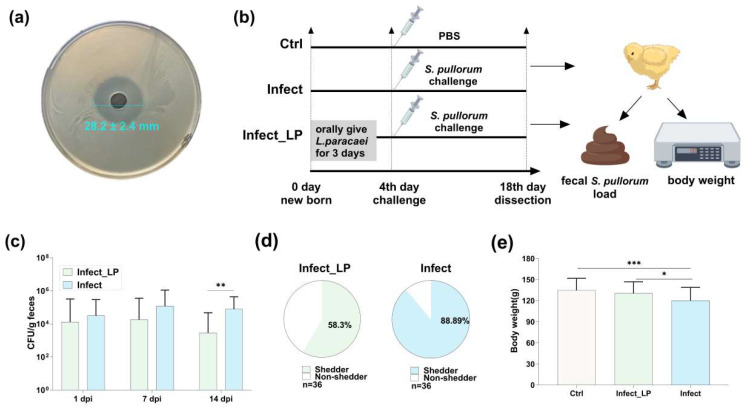
*L. paracasei* inhibits *S.* Pullorum in vitro and alleviates *S.* Pullorum infection in vivo. (**a**) The antibacterial ability of *L. paracasei* Meng cell-free supernatant measured by the liquid dilution method. (**b**) Schematic of the *L. paracasei* Meng intervention experiment. (**c**) *S.* Pullorum fecal shedding in infected chickens treated (group Infect_LP) and untreated (group Infect) with *L. paracasei* Meng. (**d**) The proportion of *S.* Pullorum shedders in infected chickens under different interventions. (**e**) Body weight variations in chickens with different treatments at 14 d.p.i. *: *p* < 0.05, **: *p* < 0.01 and ***: *p* < 0.001.

**Figure 4 microorganisms-12-01743-f004:**
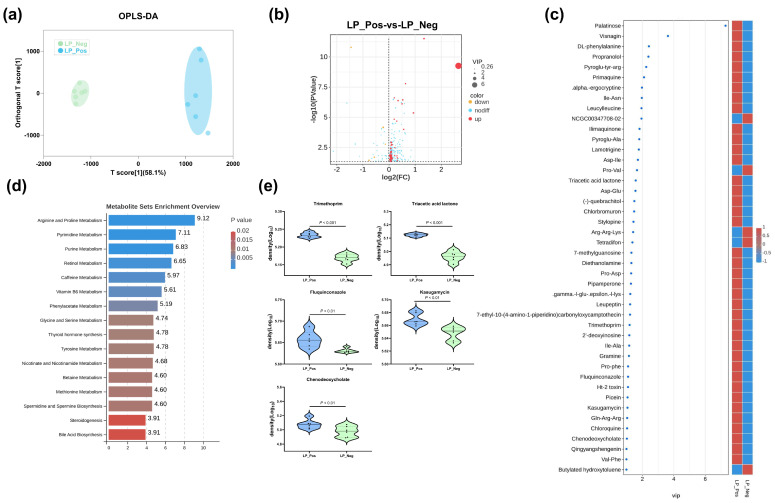
Comparative metabolomic analysis of sterile supernatants from two *L. paracasei* strains. (**a**) OPLA-DA analysis reveals different metabolite profiles between LP_Pos and LP_Neg. (**b**) Volcano plot of the metabolite regulation in LP_Pos and LP_Neg, with significantly downregulated metabolites in blue and upregulated metabolites in red (*p* < 0.05, fold change >1); the dot size represents the VIP value. (**c**) Heatmap of the regulation of 44 metabolites between two groups. Color scale: red, upregulated metabolites; blue, downregulated metabolites. The metabolite is NCGC00347708-02, which IUPAC name is *Spirostan-3-yl 6-O-β-D-xylopyranosyl-β-D-galactopyranoside*. (**d**) Heatmap of MSEA for metabolic pathway enrichment in the regulation of 44 metabolites. Abscissae represent the enrichment score (ES). Higher ES values indicate a stronger enrichment of the metabolite set. (**e**) Differential contents of six key metabolites.

**Table 1 microorganisms-12-01743-t001:** Gut microbiota with significant differences between *S.* Pullorum-infected and non-infected chickens at the genus level.

	Mean (%)		
Genus	SP	Ctrl	FC (SP/Ctrl)
*Escherichia-Shigella*	65.0	40.6	1.6 *** ↑
*Lactobacillus*	9.86	32.1	0.31 ** ↓
*Enterococcus*	10.8	3.70	2.92 *** ↑
*Ruminococcus*	0.72	1.38	0.52 * ↓
*Streptococcus*	0.16	0.83	0.19 **↓
*Bifidobacterium*	0.02	0.66	0.03 ***↓
*Ralstonia*	0.14	0.47	0.29 **↓
*Clostridium*	0.21	0.57	0.37 * ↓
*Butyricicoccus*	0.15	0.43	0.35 * ↓
*Coprobacillus*	0.05	0.41	0.12 **↓
*Caulobacter*	0.13	0.33	0.39* ↓
*Lactococcus*	0.12	0.26	0.46 * ↓
*Blautia*	0.04	0.16	0.25 **↓
*Veillonella*	0.04	0.14	0.29 **↓

* *p* < 0.05, ** *p* < 0.01 and *** *p* < 0.001. The upward arrow (↑) indicates an increase in abundance within the infected group, and the downward arrow (↓) indicates a decrease in abundance.

## Data Availability

The datasets generated during the current study are available in MG-RAST (https://www.mg-rast.org/mgmain.html?mgpage=project&project=mgp100169 (accessed on 20 June 2024)).

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
