# Peer review of "Microbiome and Microbial Pure Culture Study Reveal Commensal Microorganisms Alleviate Salmonella enterica Serovar Pullorum Infection in Chickens"

_microorganisms, 2024, doi:10.3390/microorganisms12091743_

Round 1

Reviewer 1 Report

Comments and Suggestions for Authors

The objective of this paper was to identify and evaluate the interaction between  Salmonella Pullorum and chicken gut microbiota. It is well established that Salmonella strains, particularly those isolated from the environment, chicken or food, exhibit distinct characteristics that differ from those of classical reference strains. it would be interesting to use a field strain for challenge. The paper is interesting but lack of well-defined aim of the study.

Authors focused on Salmonella enterica, nomenclature for this bacteria should be correct. Salmonella enterica enterica serovar Gallinarum biovar Pullorum (S Pullorum) or Salmonella enterica serovar Pullorum (S. Pullorum) or S. Pullorum

S. Pullorum challenge and sampling

Lack of information about bacterial and viral infection, same healthy status. Moreover, please add details about methods of checking that chickens were negative for Salmonella Pullorum and other Salmonella.

Standard strain Salmonella Pullorum 1218 were originally obtained from which collection or it was field isolate?

Line 94 - new born chicks - what breed/race?

Line 95 – suspension of bacteria was prepared in which resolvent?

16S rRNA gene amplicon sequencing

Why 60 and 48 fecal samples were collected from infected and uninfected chicken respectively, rather than the same number of samples?

S. Pullorum infection related microorganisms isolation and identification

Please use correct form of method description, like in other paragraphs.

For all bacteria only one temperature and incubation time were used (37°C and incubate for 18-24 h), but sob fastidious bacteria needs different condition.

Line 139 - agar plates, please add agar medium name

Line 141 – please add details to obtain anaerobic conditions; anaerobic tanks with sachets?

Line 151 - Supplementary Figure 2a and b should be placed in Supplementary files, not in main text. Moreover this figure describe next paragraph, Antimicrobial Test

Line 169 – Why only Lacticaseibacillus paracasei Lp02 were used to S. Pullorum challenge?

Line 170 - specific pathogen free chicken, White Leghorn breed were obtained from?

Add MRS and other media producer.

The authors are encouraged to add a schematic illustration, especially Salmonella challenge, presenting the steps conducted in this study to facilitate the following of the current investigations.

Line 213 - An animal infection model was established by inoculating S. Pullorum into the legs of chicks to study the gut microbiota – sentence should be revised

Line 216 - with losses of 14.2 g, 17.3 g, and 17.4 g – it was average for each group? Please add control group weight

Line 220 –Supplementary Figure. 1a, b, c should be in Supplement files, not in main text. In methods lack of details about S. Pullorum culture from organs, please add to methods.

Line 223 - grayishwhite nodules, indicative of pullorum disease infection – it is overinterpretation, similar symptoms occur in other diseases

Line 301 - Supplementary Table 1. Should be in Supplement files, not in main text.

 Figure 4 c, d, e is unreadable

Supplementary Figure 3. Should be in Supplement files, not in main text.

Almost half of the cited articles are outdated. 32 from 51 older than 5 years

References should be described as follows, depending on the type of work:

Journal Articles:
1. Author 1, A.B.; Author 2, C.D. Title of the article. Abbreviated Journal Name YearVolume, page range.

Author Response

Dear Reviewer:

I sincerely appreciate the time and effort you have dedicated to reviewing my manuscript. I have carefully considered your insightful comments and have responded and made amendments point by point. During the revision process, I have highlighted all the revisions made to the manuscript in red font to facilitate clear visibility of the changes. For each comment, corresponding adjustments and improvements have been made in the revised manuscript, and the detailed explanations are as follows:

Comments 1: Authors focused on Salmonella enterica, nomenclature for this bacteria should be correct. Salmonella enterica enterica serovar Gallinarum biovar Pullorum (S Pullorum) or Salmonella enterica serovar Pullorum (S. Pullorum) or S. Pullorum Response 1: Thank you for professionally pointing out the nomenclature issue regarding Salmonella enterica. We have made the necessary corrections in the revised manuscript and now use the correct nomenclature of Salmonella enterica serovar Pullorum (S. Pullorum) or S. Pullorum throughout.

Comments 2: S. Pullorum challenge and sampling. Lack of information about bacterial and viral infection, same healthy status. Moreover, please add details about methods of checking that chickens were negative.

Response 2: The S. Pullorum used in this study is an isolated strain from Shanghai Academy of Agricultural Science, and its physiological and biochemical characteristics have been reported in articles (doi: 10.3389/fcimb.2021.759965 and doi:10.3389/fvets.2021.683853), We have added the reference articles in the new version of the manuscript. Regarding the determination of whether the chickens were positive or negative for pullorum disease, we based it on the whole blood plate agglutination test. We have added the methods in the new version of the manuscript. The specific operation is as follows: Absorb 1 drop (approximately 0.05 ml) of the antigen and drip it vertically onto the glass plate. Then, take an equal amount of the blood to be tested from the chicken (by pricking the brachial vein or the tip of the comb with a needle) or the serum and mix it evenly with the antigen to spread into a circular liquid surface with a diameter of approximately 2 cm. At the same time, set up strong positive, weak positive, and negative serum controls. Determine the results within 2 minutes. When 100% agglutination ( ++++) occurs in the strong positive serum, and 50% agglutination (++) or no agglutination (-) occurs in the negative and weak positive sera, the test is valid. If more than 50% agglutination occurs in the blood (serum) to be tested, it is positive; if no agglutination occurs in the blood (serum) to be tested, it is negative; and if it is between the above two, it is judged as suspicious.

Comments 3: for Salmonella Pullorum and other Salmonella. Standard strain Salmonella Pullorum 1218 were originally obtained from which collection or it was field isolate?

Response 3: As mentioned in Response 2, the strain Salmonella Pullorum 1218 is an isolated strain from Shanghai Academy of Agricultural Science, and its physiological and biochemical characteristics have been reported in articles. We measured the lethal lethality of this strain of Salmonella Pullorum and observed adverse symptoms in chicks after challenge, the strain could be successfully used to make the challenge model.

Comments 4: Line 94 - new born chicks - what breed/race?

Response 4: They are Xin Pudong chickens, a shanghai native chicken breed. We have made the correction in the new version of the manuscript.

Comments 5: Line 95 – suspension of bacteria was prepared in which resolvent?

Response 5: We cultivated Salmonella pullorum overnight using LB (Luria-Bertani) medium. Subsequently, the bacteria were centrifuged and then resuspended with PBS to adjust to the challenge concentration. We have made the correction in the new version of the manuscript.

Comments 6: 16S rRNA gene amplicon sequencing Why 60 and 48 fecal samples were collected from infected and uninfected chicken respectively, rather than the same number of samples?

Response 6: The number of chickens in the challenge group was 353, located in 21 cages, 10 cages were randomly selected during sampling, and 6 samples were collected from each cage. In the unchallenged group, the number of chickens was 151, raised in 9 cages, 8 cages were randomly selected during sampling, and 6 samples were collected from each cage. Details are available in Supplementary Figure 1 b in the new version of the manuscript.

Comments 7: S. Pullorum infection related microorganisms isolation and identification Please use the correct form of method description, like in other paragraphs.

Response 7: We have made the correction in the new version of the manuscript.

Comments 8: For all bacteria only one temperature and incubation time were used (37°C and incubate for 18-24 h), but some fastidious bacteria need different conditions.

Response 8: Your concerns are correct, some fastidious bacteria need different conditions. We only roughly selected the temperature and time when conducting the culture, and as we mentioned in the discussion section of this question, the culture conditions are one of the reasons that limit many microbes in feces as difficult to culture.

Comments 9: Line 139 - agar plates, please add the agar medium name. 

Response 9: It is the name of the corresponding medium. For example, for the selective culture of Lactobacillus bacteria, we used MRS medium. We have made the correction in the new version of the manuscript.

Comments 10: Line 141 – please add details to obtain anaerobic conditions; anaerobic tanks with sachets?

Response 10: Yes, we use the anaerobic tank, add the deaerator to remove the oxygen from the tank. We added the reference to the culture method. 

Comments 11: Line 151 - Supplementary Figure 2a and b should be placed in Supplementary files, not in the main text. Moreover, this figure describes the next paragraph

Response 11: Thank you for your careful reminder. We have adjusted the figure.

Comments 12: Line 169 – Antimicrobial Test, Why only Lacticaseibacillus paracasei Lp02 was used for the S. Pullorum challenge?

Response 12: Before the strain was used for the S. Pullorum challenge, its inhibitory effect on Salmonella Pullorum was detected by the agar diffusion method. Compared with other strains, this Lacticaseibacillus paracasei Lp02 strain showed the best inhibitory effect on Salmonella Pullorum in the in vitro experiments. Therefore, it was used for the S. Pullorum challenge-intervention experiment .

Comments 13: Line 170 - specific pathogen free chicken, White Leghorn breed were obtained from?

Response 13: The SPF chicken was purchased from Jinan SAIS Poultry CO., LTD. We have made the revision and marked it in red font.

Comments 14: Add MRS and other media producer.

Response 14: These mentioned media were produced by Hope Bio-Technology Co., Ltd. We have made the correction in the new version of the manuscript.

Comments 15: The authors are encouraged to add a schematic illustration, especially Salmonella challenge, presenting the steps conducted in this study to facilitate the following of the current investigations.

Response 15: We have summarized the process of the challenge by drawing a diagram. For details, see supplementary figure 1 in the new version of the manuscript.

Comments 16: Line 213 - An animal infection model was established by inoculating S. Pullorum into the legs of chicks to study the gut microbiota – the sentence should be revised

Response 16: We have made the revision and marked it in red font.

Comments 17: Line 216 - with losses of 14.2 g, 17.3 g, and 17.4 g – was it the average for each group? Please add the control group weight

Response 17: With average losses of 14.2 g, 17.3 g, and 17.4 g, respectively, compared to the non-infected chickens (group Ctrl), which were 109.5 g, 154.7 g, and 171.2 g respectively (Figure. 1a). and we have made the correction in the new version of the manuscript.

Comments 18: Line 220 – Supplementary Figure. 1a, b, c should be in Supplement files, not in the main text. In the methods, there is a lack of details about S. Pullorum culture from organs. Please add to the methods.

Response 18: We have adjusted the order of the figures and we have added the culture method and conditions of S. Pullorum in the Materials and Methods section. For the detection of S. Pullorum in feces and organs, we used small steel balls to shake and grind these samples into liquids, and then cultivated S. Pullorum using the selective medium BSA.

Comments 19: Line 223 - grayishwhite nodules, indicative of pullorum disease infection – it is overinterpretation. Similar symptoms occur in other diseases Thank you for your meticulous observation.

Response 19: Grayishwhite nodules are one of the symptoms of many diseases. And grayishwhite nodules are also one of the symptoms of Salmonella pullorum infection. Here, it is only a supplement to test whether the model is successful. As described in the main text, detecting the load of Salmonella pullorum is the most important indicator, and grayishwhite nodules are an auxiliary evidence.

Comments 20: Comments 21: Line 301 - Supplementary Table 1. Should be in Supplement files, not in the main text.

Response 20: We have made the correction in the new version of the manuscript.

Comments 21: Figure 4 c, d, e is unreadable

Response 21:We adjusted the resolution of the picture for more clarity.

Comments 22: Supplementary Figure 3. Should be in Supplement files, not in the main text.

Response 22:Thank you for your careful reminder. We have adjusted the figure.

Comments 23: Almost half of the cited articles are outdated. 32 from 51 are older than 5 years

Response 23:Based on the similarity of the topic content, we have read the updated literature and updated the references to have a better grasp of this field. We have made the correction in the new version of the manuscript and we have retained some of the classic important literature.

Comments 24:References should be described as follows, depending on the type of work:

Journal Articles:
1. Author 1, A.B.; Author 2, C.D. Title of the article. Abbreviated Journal Name YearVolume, page range.

Response 24:We have made the correction in the new version of the manuscript.

Reviewer 2 Report

Comments and Suggestions for Authors

Introduction

Should be rewritten in some parts to avoid copying the whole sentences from the abstract, i.e. lines 18-20 and 56-58.

Materials and Methods

In paragraph” S. pullorum infection related microorganisms isolation and identification” the text changes its narration (looks like instruction now). This should be corrected and adjusted to the other parts of the paper.

Line 64, 67 – Clostridium difficile was reclassified in 2016, the authors are free to use the older name Clostridium difficile but I recommend to reconsider using Clostridioides difficile (when given for the second time can be C. difficile)

Line 242 – dot?

Line 260, 262 – Clostridium enumerated twice

There is no information what happened to the animals that survived after the experiment and what was the procedure.

Discussion

The first paragraph (lines 348 352) is the content of conclusions. It should be deleted.

References

33 of 55 references are older then 5 years. I suggest to update the reviewed and cited papers as studies on poultry diseases especially in the field of foodborne diseases, gut microbiota and substitutes to antibiotic therapies are highly developed

Author Response

Dear Reviewer:

I sincerely appreciate the time and effort you have dedicated to reviewing my manuscript. I have carefully considered your insightful comments and have responded and made amendments point by point. During the revision process, I have highlighted all the revisions made to the manuscript in red font to facilitate clear visibility of the changes. For each comment, corresponding adjustments and improvements have been made in the revised manuscript, and the detailed explanations are as follows:

Comments 1: Introduction Should be rewritten in some parts to avoid copying the whole sentences from the abstract, i.e. lines 18-20 and 56-58.

Response 1: Thank you for your professional advice. We have revised it in the new version of the manuscript.

Comments 2: Materials and Methods

In paragraph” S. pullorum infection related microorganisms isolation and identification” the text changes its narration (looks like instruction now). This should be corrected and adjusted to the other parts of the paper.

Response 2: This paragraph is not well written, thank you for pointing it out. We modified to summarize the methods for isolated culture of the microorganisms, and we have revised it in the new version of the manuscript.

Comments 3: Line 64, 67 – Clostridium difficile was reclassified in 2016, the authors are free to use the older name Clostridium difficile but I recommend to reconsider using Clostridioides difficile (when given for the second time can be C. difficile)

Response 3: Thank you for your professional advice. Our new version of the manuscript was revised.

Comments 4: Line 242 – dot?

Response 4: We didn't notice it, and it caused your misunderstanding. Line 242 is the row occupied by the graph.

Comments 5: Line 260, 262 – Clostridium enumerated twice

Response 5: Thank you for your careful reminder, we have made the correction in the new version of the manuscript.

Comments 6: There is no information what happened to the animals that survived after the experiment and what was the procedure.

Response 6: We draw a schematic diagram of the challenge in the new version of the manuscript, detailed on the challenge process, it can be found in Supplementary Figure 1. After the trial, the challenge group showed many bad symptoms, such as listlessness and poor appetite, and in this study, because death and survival were the most obvious manifestations, more attention was paid to death and survival.

Comments 7: Discussion

The first paragraph (lines 348-352) is the content of conclusions. It should be deleted.

Response 7: Thank you for your suggestion, we have deleted it in the new version of the manuscript.

Comments 8: References

33 of 55 references are older then 5 years. I suggest to update the reviewed and cited papers as studies on poultry diseases especially in the field of foodborne diseases, gut microbiota and substitutes to antibiotic therapies are highly developed.

Response 8: Based on the similarity of the topic content, we have read the updated literature and updated the references on poultry diseases, gut microbiota and alternative to antibiotic therapies. We have made the correction in the new version of the manuscript and we have retained some of the classic important literature.